# Critical timing: Impact of delays to surgery on prognosis in stage I-II non-small cell lung cancer

**Ye Zhang[1], Yeji Hu[1], Jinfeng Xi[1], Bo Wu[2], Wenxiong Zhang[2], Chunling Li[3]***

**1** Department of Thoracic Surgery, iangxi Provincial People's Hospital, The First Affiliated Hospital of Nanchang Medical College, Nanchang, China, **2** Department of Thoracic Surgery, The Second Affiliated Hospital, Jiangxi Medical College, Nanchang University, Nanchang, China, **3** Operating Room, iangxi Provincial People's Hospital, The First Affiliated Hospital of Nanchang Medical College, Nanchang, China

* 13807057470@163.com

## Abstract

### Background

Delaying surgery affects the prognosis of patients with lung cancer, but the critical time point at which it becomes detrimental to survival. Identifying this critical time point may benefit patients and guide clinical practice.

### Methods

Data from patients diagnosed with stage I-II non-small cell lung cancer (NSCLC) were extracted from the Surveillance, Epidemiology, and End Results (SEER) database. Univariate and multivariate Cox regression analyses were employed to evaluate prognostic factors associated with overall survival (OS) and to identify time points from diagnosis to surgery that significantly impact prognosis. Kaplan-Meier curves and subgroup analyses were conducted to validate the affect of early versus late surgery on OS. Multinomial logistic regression was utilized to evaluate factors associated with delays in the time from diagnosis to surgery.

### Results

We included 55,582 adult patients with stage I-II NSCLC from the SEER database. Time to surgery (TTS) was identified as an independent prognostic factor for OS in stage I-II NSCLC patients through multivariate Cox regression analysis. Compared to surgeries performed within 6 weeks of TTS, those performed after 6 weeks of TTS (HR: 1.22, 95% CI: 1.20–1.25, P < 0.001) were significantly related to poorer OS. Multinomial logistic regression revealed that age, sex, race, and marital status were risk factors for delayed TTS after diagnosis. Compared to patients with a TTS of 0–40 days, those with a TTS of 63–111 days had the following risks: for patients aged ≥ 75 years, the odds ratio (OR) was 1.46 (95% CI: 1.32–1.62, P < 0.001); for males, the OR was 1.15 (95% CI: 1.09–1.20, P < 0.001).

**Data availability statement:** The original data used in this study were sourced from the "SEER Research Data, 17 Registries" dataset within SEERStat 8.4.4 software (https://seer.cancer.gov/). The analyzed data are provided within the paper and its Supporting Information files.

**Funding:** The author(s) received no specific funding for this work.

**Competing interests:** The authors have declared that no competing interests exist.

**Abbreviations:** CI, confidence interval; HR, Hazard ratio; IPTW, inverse probability of treatment weighting; LSCC, lung squamous cell carcinoma; LUAD, lung adenocarcinoma; NSCLC, non-small cell lung cancer; OR, odds ratio; OS, overall survival; SEER, surveillance epidemiology and end results; TTS, time to surgery.

## Conclusion

Compared to stage I-II NSCLC patients who underwent surgery more than 6 weeks after diagnosis, those who underwent surgery within 6 weeks had significantly higher survival rates. Delays in surgery were associated with adverse social factors.

## Introduction

Lung cancer continues to be the primary cause of malignant tumors worldwide, both in terms of incidence and mortality [1]. Non-small cell lung cancer (NSCLC) constitutes the predominant histological subtype of lung cancer [2]. Surgery is regarded as the standard treatment for stage I-II NSCLC [3]. Several previous studies have found that patients with early NSCLC who experience a longer time to surgery (TTS) from diagnosis tend to exhibit a poorer prognosis [4–6]. However, it remains unclear at which specific time point a delay in surgery negatively impacts overall survival (OS) for patients with stage I-II NSCLC. Identifying this time point is crucial for enhancing prognostic outcomes related to surgical treatment. The National Comprehensive Cancer Network's guidelines recommend that delays in the time from diagnosis to surgery should be minimized [7]. However, this issue has not been further explored. In contrast, the European Consensus Conference on Lung Cancer [8] and the American College of Chest Physicians [9] do not specify the time from diagnosis within which recommendations for surgical intervention should be given. Therefore, we urgently need to clarify the acceptable waiting period between diagnosis and surgery, which will help guide patients to receive treatment promptly and surgeons to ensure the best treatment strategy.

However, it is also important to consider the factors that contribute to delays in the procedure that extend beyond an acceptable timeframe. First, surgical treatment planning typically involves the completion of additional imaging, physical status assessments, and consultations with radiologists, surgeons, and oncologists [10,11]. However, the examinations and multidisciplinary consultation time can be prolonged. These steps are necessary, as they facilitate tailoring the precise type of surgery and course of treatment to the patient. Second, delays in TTS may be related to the patient's poorer physical condition, which could also correlate with a poor prognosis for these patients [12]. Finally, various social factors also influence TTS, including whether the patient possesses health insurance and its type [13], financial conditions, family care and support, and the availability of local healthcare resources [14,15]. These factors can contribute to delays in surgical treatment. Understanding these factors can enhance healthcare delivery and may potentially improve survival rates.

This study assessed the relationship between TTS and OS in stage I-II NSCLC patients utilizing the Surveillance, Epidemiology, and End Results (SEER) database. It aims to determine the association between TTS delays and survival outcomes, while establishing an optimal surgical window for clinical decision-making. Additionally, we analyzed the factors contributing to delayed TTS.

## Methods

### Data source

This study focused on patients diagnosed with stage I-II NSCLC between 2000 and 2016, utilizing data from the SEER database, with a follow-up period extending to 2021. The SEER database, established by the National Cancer Institute in the United States, is an extensive oncology resource that collected data on the incidence, mortality, and prevalence of malignant tumors among millions of patients in various selected states and counties across the country [16]. Since the SEER data is publicly accessible, this study was exempt from obtaining approval from the hospital's Ethics Review Committee.

### Inclusion and exclusion criteria

Adult patients (≥ 18 years) with a pathological diagnosis of stage I-II NSCLC recorded in the SEER database from 2000 to 2016 were enrolled in this study. Patients who did not undergo surgery or whose surgical status was unknown were excluded from the study. Additionally, patients receiving non-surgical treatments, such as radiotherapy or chemotherapy, were excluded. Finally, patients with a TTS of 0 days were also excluded, as these cases represented excisional biopsies or incidental surgical findings of NSCLC. The screening process for the inclusion of patients is detailed in Fig 1.

### Study variables

Demographic and clinical variables included year of diagnosis, age, sex, race, marital status, household income, rural-urban county of residence, histologic type, grade, lymph node status, tumor size, stage, surgery type and TTS. TTS was classified into intervals of 0–4, 5, 6, 7, 8, 9, 10, 11, and ≥ 12 weeks. The 1-month TTS range is the threshold used in prior literature [17], and by the British Thoracic Society [18]. Therefore, the choice of our 0–4 weeks (1–28days) TTS subgroup is clinically justified. Weekly intervals were used after the first month to gain a more nuanced understanding of

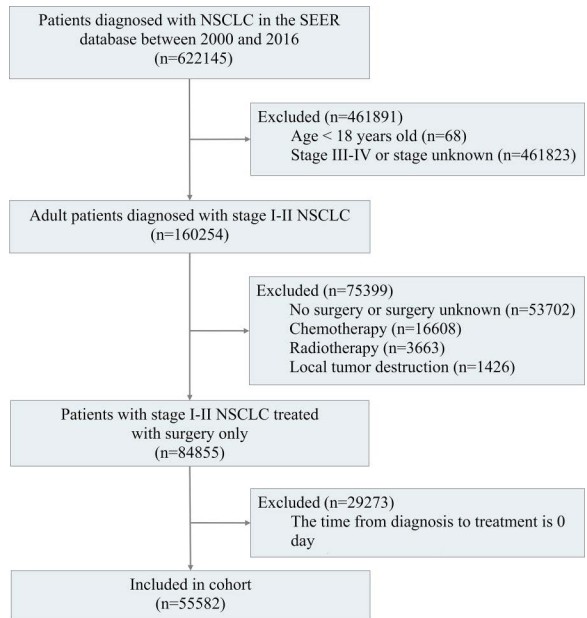

**Fig 1. The flowchart of stage I-II NSCLC inclusion population.**

the relationship between TTS and OS. The primary predictor variable was the time from diagnosis to the first surgery for NSCLC patients, with OS serving as the outcome metric.

## Statistical analysis

Descriptive statistics were used to describe the characteristics of the population included in the study. Multiple methods were utilized to determine the impact of prolonged TTS on survival. First, independent prognostic factors related to OS in patients with stage I-II NSCLC were identified by univariate and multivariate Cox regression analyses. TTS thresholds were determined according to the Cox regression results, identifying the time points with significant differences in survival compared to early surgery. Furthermore, Kaplan-Meier curves were generated to evaluate the impact of TTS on patient survivability. Subsequently, patients were categorized into early and late surgery groups based on the established TTS thresholds. Inverse probability of treatment weighting (IPTW) was used to balance baseline characteristics between early and late surgery groups. In the subsequent analysis, TTS as a time-dependent covariate to avoid immortal time biases. Comparison of OS between the early surgery group and the late surgery group by univariate and multivariate Cox regression analyses after IPTW, with Kaplan-Meier curves again utilized to validate differences in survivability. Finally, subgroup analyses were conducted to determine the applicability of the findings across various subgroups.

Additionally, multinomial logistic regression was conducted to identify clinical characteristics linked to a longer duration of TTS [19]. A new categorical variable based on TTS was created using TTS = 40 days after diagnosis (50th percentile) as the reference. Comparison time groups included 41–62 days (51–75th percentile patients), 63–111 days (76–95th percentile), and > 112 days (> 95th percentile). This approach allows for a precise analysis of the relative importance of various factors influencing delayed TTS.

Statistical analyses were performed utilizing R Language software version 4.3.2 (Vienna, Austria) and RStudio software version 2024.04.02. The baseline characteristics of variables between groups were compared using the Pearson chi-square test, with p-values below 0.05 considered statistically significant.

## Results

### Patient characteristics

622,145 patients diagnosed with NSCLC from 2000 to 2016 were retrieved from the SEER database, of which 160,254 were adults with stage I-II NSCLC. We then excluded patients with no surgery or unknown surgical status (n = 53,702), those who received radiotherapy (n = 16,608), chemotherapy (n = 3,663), local tumor destruction (n = 1,426) and those with a TTS of 0 days (n = 29273). Ultimately, our study included 55,582 stage I-II NSCLC patients (Fig 1). These demographic characteristics are detailed in Table 1, 39763 (71.49%) patients were aged over 65 years old, 28,211 (50.76%) patients were female and 47,567 (85.58%) patients were White. The majority of patients underwent lobectomy (n = 44,562, 80.17%), while 6,999 (12.59%) patients underwent pulmonary wedge resection, and 2,149 (3.87%) underwent segmental resection. The remaining 1,872 (3.37%) patients underwent other pulmonary procedures. The median (IQR) TTS was 40 (25–62) days, of whom 27,975 patients were operated on within 40 days. A total of 41,698 patients (75th percentile) were operated on within 62 days (Fig 2).

### Survival analysis

Univariate Cox regression demonstrated that all variables are prognostic factors for OS in patients with stage I-II NSCLC. Furthermore, multivariate Cox regression analyses showed that an increase in TTS was associated with a poorer prognosis (Table 2, Fig S1). Notably, no statistically significant difference in OS was observed among patients in any surgical group who experienced TTS within 6 weeks (P > 0.05). OS was significantly worse in the group with a TTS of 7 weeks compared to the earliest surgery group (HR: 1.08, 95% CI: 1.04–1.12, P < 0.001). Additionally, Kaplan-Meier curves demonstrated results consistent with these findings (Fig 3A). As anticipated, diagnosis during 2000–2005, advanced age, male gender, Black ethnicity, lower income, lack of family support (marital status is divorced, single and

**Table 1. Cohort characteristics.**

| Demographic characteristics | Cohort, No (%) (N = 55,582) |
|---|---|
| **Year of diagnosis** | |
| 2000-2005 | 16682 (30.01%) |
| 2006-2011 | 21356 (38.42%) |
| 2012-2016 | 17544 (31.56%) |
| **Age** | |
| ≤ 55 years | 3987 (7.17%) |
| 55–64 years | 11859 (21.34%) |
| 65–74 years | 21901 (39.4%) |
| ≥ 75 years | 17835 (32.09%) |
| **Sex** | |
| Female | 28211 (50.76%) |
| Male | 27371 (49.24%) |
| **Race** | |
| White | 47567 (85.58%) |
| Black | 4232 (7.61%) |
| Other | 3718 (6.69%) |
| Unknown | 65 (0.12%) |
| **Marital status** | |
| Married | 31460 (56.6%) |
| Divorced | 7027 (12.64%) |
| Single (never married) | 5658 (10.18%) |
| Widow | 9642 (17.35%) |
| Unknown | 1795 (3.23%) |
| **Household income** | |
| ≤ $54999 | 7171 (12.9%) |
| $55,000 - $74,999 | 21064 (37.9%) |
| $75,000 - $94,999 | 15662 (28.18%) |
| > $94,999 | 11685 (21.02%) |
| **Rural-urban county of residence** | |
| Metropolitan (≥ 1 million populations) | 31432 (56.55%) |
| Metropolitan (25000–1 million populations) | 11503 (20.7%) |
| Metropolitan (< 25000 populations) | 4768 (8.58%) |
| Urban | 4618 (8.31%) |
| Rural | 3196 (5.75%) |
| Unknown | 65 (0.12%) |
| **Histologic type** | |
| LUAD | 33558 (60.38%) |
| LSCC | 16553 (29.78%) |
| Other | 3759 (6.76%) |
| Unknown | 1712 (3.08%) |
| **Grade** | |
| I | 8412 (15.13%) |
| II | 23779 (42.78%) |
| III | 18608 (33.48%) |
| IV | 1142 (2.05%) |
| Unknown | 3641 (6.55%) |

*(Continued)*

Table 1. (Continued)

| Demographic characteristics | Cohort, No (%) (N = 55,582) |
| --- | --- |
| **Lymph node positivity** | |
| 0 | 46096 (82.93%) |
| 1-3 | 3623 (6.52%) |
| >3 | 490 (0.88%) |
| Unknown | 5373 (9.67%) |
| **Tumor size** | |
| 0–3 cm | 37070 (66.69%) |
| >2 and ≤ 5 cm | 13248 (23.84%) |
| >5 cm | 4822 (8.68%) |
| Unknown | 442 (0.8%) |
| **Stage** | |
| I | 49927 (89.83%) |
| II | 5655 (10.17%) |
| **Surgery type** | |
| Wedge resection | 6999 (12.59%) |
| Segmentectomy | 2149 (3.87%) |
| Lobectomy | 44562 (80.17%) |
| Other | 1872 (3.37%) |
| **Time to surgery** | |
| 0–4 Weeks (1–28 days) | 17541 (31.56%) |
| 5 Weeks (29–35 days) | 6591 (11.86%) |
| 6 Weeks (36–42 days) | 5853 (10.53%) |
| 7 Weeks (43–49 days) | 4878 (8.78%) |
| 8 Weeks (50–56 days) | 4061 (7.31%) |
| 9 Weeks (57–63 days) | 3344 (6.02%) |
| 10 Weeks (64–70 days) | 2593 (4.67%) |
| 11 Weeks (71–77 days) | 1957 (3.52%) |
| ≥ 12 Weeks (> 77 days) | 8764 (15.77%) |

Abbreviations: LSCC, lung squamous cell carcinoma; LUAD, lung adenocarcinoma.

widow), and residing in rural areas were recognized as risk factors affecting the OS of patients with stage I-II NSCLC. Regarding tumor characteristics, higher tumor grades (II: HR: 1.36, 95% CI: 1.31–1.40, III: HR: 1.47, 95% CI: 1.41–1.52, IV: HR: 1.42, 95% CI: 1.32–1.54, all $P < 0.001$), larger tumor size (> 3 and ≤ 5 cm: HR: 1.20, 95% CI, 1.17–1.23, >5 cm: HR: 1.37, 95% CI: 1.32–1.42, all $P < 0.001$) and stage II (HR, 1.52, 95% CI: 1.43–1.61, $P < 0.001$) are identified as risk factors affecting the OS of patients with stage I-II NSCLC (Table 2). Patients were categorized into early surgery (TTS ≤ 6 weeks) and late surgery (TTS > 6 weeks) groups based on a TTS of 6 weeks as the threshold. The baseline characteristics of the two groups before and after IPTW are in Table S1. The results of IPTW-adjusted univariate and multivariate Cox regression analyses and the corresponding forest plots are detailed in Table 3 and Fig S2. Patients with TTS ≤ 6 weeks group had a better prognosis than those with TTS > 6 weeks after IPTW (HR: 1.22, 95% CI: 1.20–1.25, $P < 0.001$) (Table 3). These findings were supported by Kaplan-Meier curves (Fig 3B, C). Additionally, Kaplan-Meier curves subgroup analysis showed that patients who underwent surgery within 6 weeks had a significantly better prognosis than patients who had late surgery in all subgroups, except for the subgroups of Black patients, lymph node positivity (> 3) and grade IV (Fig 4, Fig S3-S4).

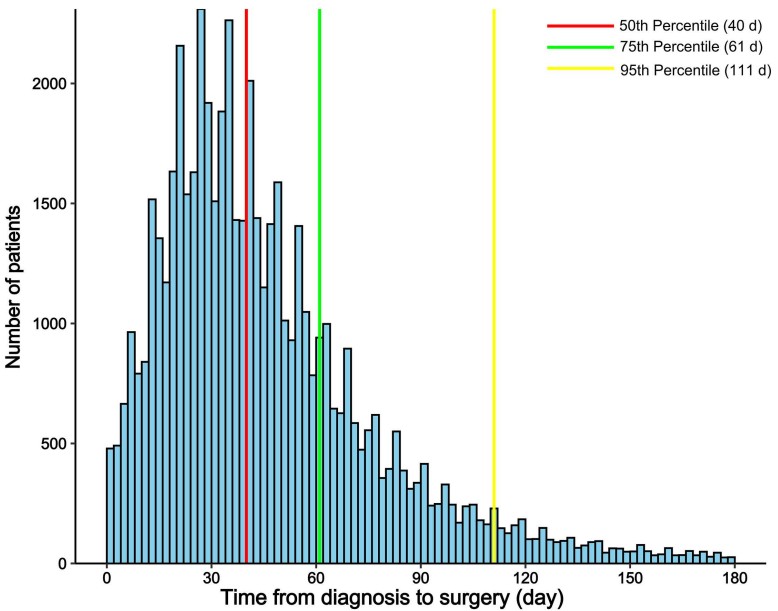

**Fig 2. Histogram of time from diagnosis to surgery and stacked plot of percentage of patients undergoing surgery at each time point.**

## Risk factors for prolonged TTS in patients with stage I-II NSCLC

Multinomial logistic regression analysis was performed to assess the risk factors associated with prolonged TTS. The analysis demonstrated that older age was associated with longer TTS (Table 4). The odds ratios (OR) for delaying TTS for patients aged ≥ 75 years, in comparison to those with TTS ≤ 40 days, were 1.48 (95% CI: 1.36–1.63) for delays of 41–62 days, 1.46 (95% CI: 1.32–1.62) for delays of 63–111 days, and 1.27 (95% CI: 1.09–1.47) for delays exceeding 111 days. Additionally, sex and race were significantly associated with prolonged TTS. The odds ratios (OR) for delaying TTS for male patients, in comparison to those with TTS ≤ 40 days, were 1.04 (95% CI: 1.00–1.09) for delays of 41–62 days, 1.15 (95% CI: 1.09–1.20) for delays of 63–111 days, and 1.32 (95% CI: 1.22–1.42) for delays exceeding 111 days. Surgical delays for Black patients are detailed in Table 4. Interestingly, marital status was significantly associated with longer TTS. Specifically, patients lacking family support had higher OR for delaying TTS to 41–62 days, 63–111 days, and > 111 days compared with patients with TTS ≤ 40 days. Household income, rural-urban county of residence and tumor characteristics were included as control variables (Table S2).

We further investigated and compared the differences in TTS among the three primary surgical approaches. Compared to the median (IQR) of TTS for lung wedge resections (43 (26, 69) days), the median (IQR) of TTS for resections of segmentectomy (41 (24, 66) days, P = 0.002) and lobectomy (40 (25, 62) days, P < 0.001) was significantly shorter. There was no significant difference between the TTS of segmentectomy and lobectomy.

## Discussion

Numerous studies have demonstrated a correlation between prolonged TTS and poorer prognosis in NSCLC [4–6], particularly during the COVID-19 outbreak [20]. However, the precise timing of delayed surgery that negatively impacts overall survival remains uncertain. Previous studies frequently measured TTS in months or categorized patients into early and late surgery groups based on median TTS, rather than using more precise units such as days or weeks [4,21]. The factors contributing to delayed TTS and their impact on survival are complex, and there remains a lack of expert consensus on the effects of surgical delay on survival [22]. Additionally, few studies have investigated the reasons for delayed TTS,

**Table 2. Univariate and multivariate Cox regression analyses for overall survival in patients undergoing primary surgery for stage I-II NSCLC.**

| Demographic characteristics | Univariate | | Multivariate | |
|---|---|---|---|---|
| | HR (95% CI) | P value | HR (95% CI) | P value |
| **Year of diagnosis** | | | | |
| 2000-2005 | 1.00 (Reference) | | 1.00 (Reference) | |
| 2006-2011 | 0.89 (0.87, 0.91) | < 0.001 | 0.90 (0.88, 0.92) | < 0.001 |
| 2012-2016 | 0.72 (0.70, 0.74) | < 0.001 | 0.75 (0.73, 0.78) | < 0.001 |
| **Age** | | | | |
| <55 years | 1.00 (Reference) | | 1.00 (Reference) | |
| 55–64 years | 1.44 (1.37, 1.52) | < 0.001 | 1.42 (1.35, 1.49) | < 0.001 |
| 65–74 years | 2.03 (1.93, 2.13) | < 0.001 | 2.00 (1.91, 2.10) | < 0.001 |
| ≥75 years | 3.06 (2.92, 3.22) | < 0.001 | 2.98 (2.83, 3.13) | < 0.001 |
| **Sex** | | | | |
| Female | 1.00 (Reference) | | 1.00 (Reference) | |
| Male | 1.45 (1.42, 1.48) | < 0.001 | 1.37 (1.34, 1.40) | < 0.001 |
| **Race** | | | | |
| White | 1.00 (Reference) | | 1.00 (Reference) | |
| Black | 0.94 (0.91, 0.98) | 0.002 | 0.99 (0.95, 1.03) | 0.560 |
| Other | 0.70 (0.67, 0.74) | < 0.001 | 0.80 (0.77, 0.84) | < 0.001 |
| Unknown | 0.32 (0.20, 0.52) | < 0.001 | 0.44 (0.28, 0.72) | 0.001 |
| **Marital status** | | | | |
| Married | 1.00 (Reference) | | 1.00 (Reference) | |
| Divorced | 1.05 (1.02, 1.08) | 0.003 | 1.19 (1.16, 1.23) | < 0.001 |
| Single (never married) | 0.97 (0.93, 1.00) | 0.048 | 1.14 (1.10, 1.18) | < 0.001 |
| Widow | 1.31 (1.27, 1.34) | < 0.001 | 1.20 (1.17, 1.24) | < 0.001 |
| Unknown | 1.07 (1.01, 1.13) | 0.021 | 1.16 (1.10, 1.23) | < 0.001 |
| **Household income** | | | | |
| ≤$54999 | 1.00 (Reference) | | 1.00 (Reference) | |
| $55,000 - $74,999 | 0.88 (0.85, 0.91) | < 0.001 | 0.88 (0.85, 0.92) | < 0.001 |
| $75,000 - $94,999 | 0.81 (0.78, 0.83) | < 0.001 | 0.84 (0.81, 0.88) | < 0.001 |
| >$94,999 | 0.79 (0.76, 0.81) | < 0.001 | 0.83 (0.79, 0.87) | < 0.001 |
| **Rural-urban county of residence** | | | | |
| Metropolitan (≥1 million populations) | 1.00 (Reference) | | 1.00 (Reference) | |
| Metropolitan (25000–1 million populations) | 1.06 (1.04, 1.09) | < 0.001 | 1.06 (1.03, 1.08) | < 0.001 |
| Metropolitan (<25000 populations) | 1.14 (1.10, 1.19) | < 0.001 | 1.07 (1.03, 1.11) | < 0.001 |
| Urban | 1.21 (1.17, 1.25) | < 0.001 | 1.07 (1.02, 1.11) | 0.004 |
| Rural | 1.23 (1.17, 1.28) | < 0.001 | 1.05 (1.00, 1.11) | 0.047 |
| Unknown | 1.20 (0.91, 1.58) | 0.202 | 1.84 (1.39, 2.44) | < 0.001 |
| **Histologic type** | | | | |
| LUAD | 1.00 (Reference) | | 1.00 (Reference) | |
| LSCC | 1.57 (1.54, 1.61) | < 0.001 | 1.18 (1.15, 1.21) | < 0.001 |
| Other | 1.49 (1.43, 1.55) | < 0.001 | 1.19 (1.14, 1.24) | < 0.001 |
| Unknown | 1.46 (1.38, 1.54) | < 0.001 | 1.14 (1.08, 1.20) | < 0.001 |
| **Grade** | | | | |
| I | 1.00 (Reference) | | 1.00 (Reference) | |
| II | 1.55 (1.50, 1.60) | < 0.001 | 1.36 (1.31, 1.40) | < 0.001 |
| III | 1.86 (1.80, 1.92) | < 0.001 | 1.47 (1.41, 1.52) | < 0.001 |
| IV | 1.96 (1.82, 2.10) | < 0.001 | 1.42 (1.32, 1.54) | < 0.001 |

*(Continued)*

**Table 2.** (Continued)

| Demographic characteristics | Univariate | | Multivariate | |
|---|---|---|---|---|
| | HR (95% CI) | P value | HR (95% CI) | P value |
| Unknown | 1.42 (1.35, 1.49) | < 0.001 | 1.23 (1.17, 1.29) | < 0.001 |
| **Lymph node positivity** | | | | |
| 0 | 1.00 (Reference) | | 1.00 (Reference) | |
| 1-3 | 1.87 (1.80, 1.94) | < 0.001 | 1.04 (0.97, 1.11) | 0.244 |
| >3 | 2.58 (2.35, 2.83) | < 0.001 | 1.44 (1.29, 1.61) | < 0.001 |
| Unknown | 1.54 (1.49, 1.58) | < 0.001 | 1.25 (1.21, 1.30) | < 0.001 |
| **Tumor size** | | | | |
| 0–3 cm | 1.00 (Reference) | | 1.00 (Reference) | |
| >3 and ≤ 5 cm | 1.35 (1.32, 1.38) | < 0.001 | 1.20 (1.17, 1.23) | < 0.001 |
| >5 cm | 1.72 (1.66, 1.77) | < 0.001 | 1.37 (1.32, 1.42) | < 0.001 |
| Unknown | 1.82 (1.64, 2.02) | < 0.001 | 1.56 (1.40, 1.73) | < 0.001 |
| **Stage** | | | | |
| I | 1.00 (Reference) | | 1.00 (Reference) | |
| II | 1.89 (1.84, 1.95) | < 0.001 | 1.52 (1.43, 1.61) | < 0.001 |
| **Surgery type** | | | | |
| Wedge resection | 1.00 (Reference) | | 1.00 (Reference) | |
| Segmentectomy | 0.85 (0.80, 0.90) | < 0.001 | 0.91 (0.86, 0.97) | 0.002 |
| Lobectomy | 0.71 (0.69, 0.73) | < 0.001 | 0.75 (0.72, 0.78) | < 0.001 |
| Other | 1.19 (1.12, 1.26) | < 0.001 | 0.91 (0.86, 0.97) | 0.005 |
| **Time to surgery** | | | | |
| 0–4 Weeks (1–28 days) | 1.00 (Reference) | | 1.00 (Reference) | |
| 5 Weeks (29–35 days) | 1.03 (1.00, 1.07) | 0.062 | 1.02 (0.99, 1.06) | 0.237 |
| 6 Weeks (36–42 days) | 1.06 (1.02, 1.10) | 0.002 | 1.04 (1.00, 1.08) | 0.056 |
| 7 Weeks (43–49 days) | 1.12 (1.07, 1.16) | < 0.001 | 1.08 (1.04, 1.12) | < 0.001 |
| 8 Weeks (50–56 days) | 1.18 (1.13, 1.23) | < 0.001 | 1.13 (1.08, 1.18) | < 0.001 |
| 9 Weeks (57–63 days) | 1.22 (1.17, 1.28) | < 0.001 | 1.16 (1.11, 1.21) | < 0.001 |
| 10 Weeks (64–70 days) | 1.24 (1.18, 1.30) | < 0.001 | 1.16 (1.10, 1.22) | < 0.001 |
| 11 Weeks (71–77 days) | 1.23 (1.17, 1.30) | < 0.001 | 1.17 (1.11, 1.24) | < 0.001 |
| ≥12 Weeks (> 77 days) | 1.29 (1.25, 1.33) | < 0.001 | 1.23 (1.19, 1.27) | < 0.001 |

**Abbreviations:** CI, confidence interval; HR, Hazard ratio; LSCC, lung squamous cell carcinoma; LUAD, lung adenocarcinoma; NSCLC, non-small cell lung cancer.

which is crucial for healthcare systems to identify and mitigate detrimental delays in treatment [23]. Therefore, it is essential to establish acceptable time frames for diagnosis and surgery while identifying factors contributing to delays in surgical procedures. In this study, we investigated whether TTS delays are associated with poorer survival outcomes and defined a safe TTS range by analyzing data from the SEER database. Furthermore, we examined factors related to TTS delays.

Consistent with the majority of prior studies, we demonstrated that TTS serves as an independent prognostic indicator for patients with stage I-II NSCLC [24]. Unlike prior studies, we assessed the impact of delayed surgery on survival using weeks as the unit of time, rather than arbitrary cut-off values or months, which are inherently flawed [4,6]. In our study, we found that delayed surgery with TTS > 6 weeks was associated with a poorer prognosis. Unfortunately, we observed that some of the included patients did not undergo timely surgical treatment. The median (IQR) TTS was 40 (25–62) days. Merely 54.95% received surgery within 6 weeks after diagnosis. With the exception of the subgroups of Black, lymph node counts of > 3, and grade IV, patients with delayed surgery had a poorer prognosis than those with earlier surgery. This

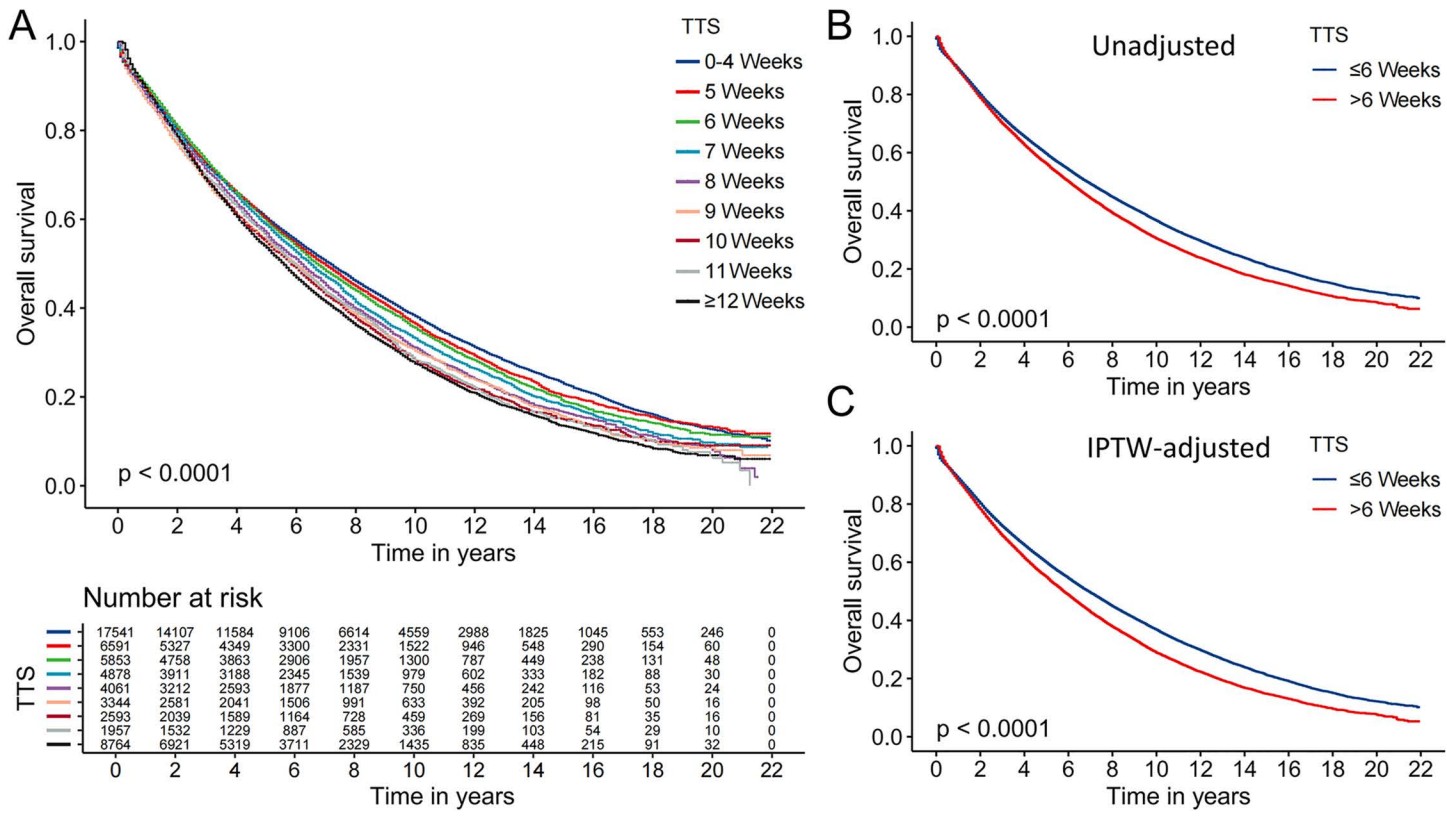

**Fig 3. Kaplan-Meier curves for overall survival of stage I-II NSCLC patients.** (A) classification by preoperative TTS: 0-4, 5, 6, 7, 8, 9, 10, 11, and ≥ 12 weeks, (B) stratification by TTS ≤ 6 weeks vs. > 6 weeks before IPTW, (C) stratification by TTS ≤ 6 weeks vs. > 6 weeks after IPTW.

underscores the accuracy and general applicability of this surgical time point. There was no significant difference in OS between timely and delayed surgery in Grade IV patients, suggesting that the biological predisposition or high mortality in patients with undifferentiated tumors may obscures survival benefit of timely surgery [25]. TTS serves as an important quality indicator for lung cancer care [26,27]. Our study further provides evidence that early-stage NSCLC should undergo surgery within 6 weeks after diagnosis. Our study holds significant clinical relevance because it supports the practice of the current healthcare model, which provides patients with adequate time to make informed decisions regarding treatment.

The second key finding of our study was to identify factors related to prolonged TTS. Although this result does not directly identify the reasons for TTS delays, it provides important insights for further investigation into the complex factors contributing to extended treatment times. First, our findings indicate that that as the year of diagnosis gets further back in time, the delay in surgery becomes more severe. This may be attributed to evolving clinical pathways requiring multidisciplinary coordination and enhanced preoperative risk stratification protocols, both of which inherently extend the time-to-treatment intervals. Second, increased age correlates with longer TTS. This could be attributed to older patients having more health issues and chronic conditions, necessitating additional preoperative assessments, such as MRI scans or cardiac evaluations [28]. Furthermore, elderly patients may also consider psychological and social factors, such as concerns about surgical risks or uncertainties regarding postoperative recovery. Second, male patients experienced significantly longer TTS. Men may be more inclined to ignore symptoms or delay seeking medical care. Consistent with the findings of

**Table 3. Univariate and multivariate Cox regression analyses of overall survival in patients undergoing primary surgery for stage I-II NSCLC after IPTW (Stratified by TTS ≤ 6 weeks and > 6 weeks).**

| Demographic characteristics | Univariate | | Multivariate | |
|---|---|---|---|---|
| | HR (95% CI) | P value | HR (95% CI) | P value |
| **Year of diagnosis** | | | | |
| 2000-2005 | 1.00 (Reference) | | 1.00 (Reference) | |
| 2006-2011 | 0.90 (0.87, 0.92) | < 0.001 | 0.91 (0.89, 0.94) | < 0.001 |
| 2012-2016 | 0.74 (0.72, 0.76) | < 0.001 | 0.77 (0.75, 0.79) | < 0.001 |
| **Age** | | | | |
| <55 years | 1.00 (Reference) | | 1.00 (Reference) | |
| 55–64 years | 1.44 (1.37, 1.52) | < 0.001 | 1.42 (1.35, 1.50) | < 0.001 |
| 65–74 years | 2.00 (1.90, 2.11) | < 0.001 | 1.99 (1.89, 2.09) | < 0.001 |
| ≥75 years | 3.01 (2.86, 3.17) | < 0.001 | 2.96 (2.81, 3.12) | < 0.001 |
| **Sex** | | | | |
| Female | 1.00 (Reference) | | 1.00 (Reference) | |
| Male | 1.45 (1.42, 1.48) | < 0.001 | 1.37 (1.34, 1.40) | < 0.001 |
| **Race** | | | | |
| White | 1.00 (Reference) | | 1.00 (Reference) | |
| Black | 0.94 (0.90, 0.97) | 0.001 | 1.00 (0.96, 1.04) | 0.919 |
| Other | 0.70 (0.67, 0.73) | < 0.001 | 0.80 (0.76, 0.84) | < 0.001 |
| Unknown | 0.36 (0.22, 0.59) | < 0.001 | 0.49 (0.30, 0.79) | 0.004 |
| **Marital status** | | | | |
| Married | 1.00 (Reference) | | 1.00 (Reference) | |
| Divorced | 1.04 (1.01, 1.07) | 0.02 | 1.19 (1.15, 1.23) | < 0.001 |
| Single (never married) | 0.96 (0.92, 0.99) | 0.017 | 1.14 (1.10, 1.18) | < 0.001 |
| Widow | 1.29 (1.26, 1.33) | < 0.001 | 1.20 (1.17, 1.24) | < 0.001 |
| Unknown | 1.06 (1.00, 1.12) | 0.06 | 1.15 (1.09, 1.23) | < 0.001 |
| **Household income** | | | | |
| ≤$54999 | 1.00 (Reference) | | 1.00 (Reference) | |
| $55,000 - $74,999 | 0.87 (0.85, 0.90) | < 0.001 | 0.88 (0.84, 0.91) | < 0.001 |
| $75,000 - $94,999 | 0.80 (0.78, 0.83) | < 0.001 | 0.84 (0.80, 0.87) | < 0.001 |
| >$94,999 | 0.79 (0.76, 0.81) | < 0.001 | 0.82 (0.79, 0.86) | < 0.001 |
| **Rural-urban county of residence** | | | | |
| Metropolitan (≥ 1 million populations) | 1.00 (Reference) | | 1.00 (Reference) | |
| Metropolitan (25000–1 million populations) | 1.07 (1.04, 1.10) | < 0.001 | 1.06 (1.03, 1.09) | < 0.001 |
| Metropolitan (< 25000 populations) | 1.15 (1.11, 1.19) | < 0.001 | 1.07 (1.03, 1.11) | 0.001 |
| Urban | 1.21 (1.17, 1.26) | < 0.001 | 1.06 (1.02, 1.11) | 0.008 |
| Rural | 1.23 (1.18, 1.29) | < 0.001 | 1.05 (1, 10.11) | 0.065 |
| Unknown | 1.19 (0.91, 1.54) | 0.201 | 1.87 (1.41, 2.49) | < 0.001 |
| **Histologic type** | | | | |
| LUAD | 1.00 (Reference) | | 1.00 (Reference) | |
| LSCC | 1.57 (1.53, 1.60) | < 0.001 | 1.18 (1.15, 1.21) | < 0.001 |
| Other | 1.49 (1.42, 1.55) | < 0.001 | 1.19 (1.14, 1.24) | < 0.001 |
| Unknown | 1.46 (1.37, 1.54) | < 0.001 | 1.15 (1.08, 1.21) | < 0.001 |
| **Grade** | | | | |
| I | 1.00 (Reference) | | 1.00 (Reference) | |
| II | 1.54 (1.49, 1.59) | < 0.001 | 1.35 (1.31, 1.40) | < 0.001 |
| III | 1.85 (1.79, 1.91) | < 0.001 | 1.46 (1.41, 1.51) | < 0.001 |

*(Continued)*

**Table 3.** (Continued)

| Demographic characteristics | Univariate | | Multivariate | |
|---|---|---|---|---|
| | HR (95% CI) | P value | HR (95% CI) | P value |
| IV | 1.94 (1.80, 2.09) | < 0.001 | 1.40 (1.30, 1.51) | < 0.001 |
| Unknown | 1.41 (1.34, 1.48) | < 0.001 | 1.23 (1.17, 1.29) | < 0.001 |
| **Lymph node positivity** | | | | |
| 0 | 1.00 (Reference) | | 1.00 (Reference) | |
| 1-3 | 1.86 (1.78, 1.94) | < 0.001 | 1.03 (0.97, 1.11) | 0.343 |
| >3 | 2.61 (2.31, 2.95) | < 0.001 | 1.44 (1.29, 1.61) | < 0.001 |
| Unknown | 1.53 (1.48, 1.58) | < 0.001 | 1.26 (1.21, 1.31) | < 0.001 |
| **Tumor size** | | | | |
| 0–3 cm | 1.00 (Reference) | | 1.00 (Reference) | |
| >3 and ≤ 5 cm | 1.35 (1.31, 1.38) | < 0.001 | 1.20 (1.17, 1.22) | < 0.001 |
| >5 cm | 1.71 (1.65, 1.78) | < 0.001 | 1.37 (1.33, 1.42) | < 0.001 |
| Unknown | 1.82 (1.62, 2.06) | < 0.001 | 1.55 (1.39, 1.72) | < 0.001 |
| **Stage** | | | | |
| I | 1.00 (Reference) | | 1.00 (Reference) | |
| II | 1.89 (1.82, 1.96) | < 0.001 | 1.53 (1.44, 1.62) | < 0.001 |
| **Surgery type** | | | | |
| Wedge resection | 1.00 (Reference) | | 1.00 (Reference) | |
| Segmentectomy | 0.85 (0.80, 0.90) | < 0.001 | 0.92 (0.86, 0.97) | 0.003 |
| Lobectomy | 0.72 (0.70, 0.74) | < 0.001 | 0.75 (0.72, 0.78) | < 0.001 |
| Other | 1.19 (1.11, 1.27) | < 0.001 | 0.91 (0.86, 0.97) | 0.006 |
| **Time to surgery** | | | | |
| ≤6 Weeks (1–42 days) | 1.00 (Reference) | | 1.00 (Reference) | |
| >6 Weeks (> 42 days) | 1.23 (1.20, 1.25) | < 0.001 | 1.22 (1.20, 1.25) | < 0.001 |

**Abbreviations:** CI, confidence interval; HR, Hazard ratio; IPTW, inverse probability of treatment weighting; LSCC, lung squamous cell carcinoma; LUAD, lung adenocarcinoma; NSCLC, non-small cell lung cancer.

Lisa R, et al. Black patients also tend to experience longer delays in surgical treatment, potentially due to racial disparities in receiving surgical recommendations and higher rates of surgery refusal among Black individuals compared to their White counterparts [29]. Addressing racism and ensuring equity in cancer interventions can help Black patients receive timely lung cancer treatment and improve OS [30]. Interestingly, patients who are divorced, single, or widowed also experienced extended TTS, possibly because married patients obtain stronger spousal support for surgery and support from their spouses [31]. Given the complex social factors affecting delays in lung cancer surgery, the healthcare system must track patients with surgery times exceeding 6 weeks to identify modifiable factors that may hinder timely surgery and negatively impact outcomes. Early intervention on these modifiable factors is crucial for reducing survival disadvantages due to surgical delays and improving patient prognosis.

In this study, we analyzed the range of acceptable TTS times based on the SEER database. The strength of our study comes from analyzing extensive tumor data samples, yielding reliable results. Secondly, we selected weekly intervals to obtain a more precise understanding of the relationship between TTS and OS. Finally, we analyzed factors associated with delayed TTS that could inform preventive strategies for patients identified as high-risk for delayed surgery. Naturally, the study has limitations. First, it is a retrospective analysis that relies on database data. Second, the SEER database lacks comprehensive information regarding significant comorbidities, which may affect the accuracy of our findings. For

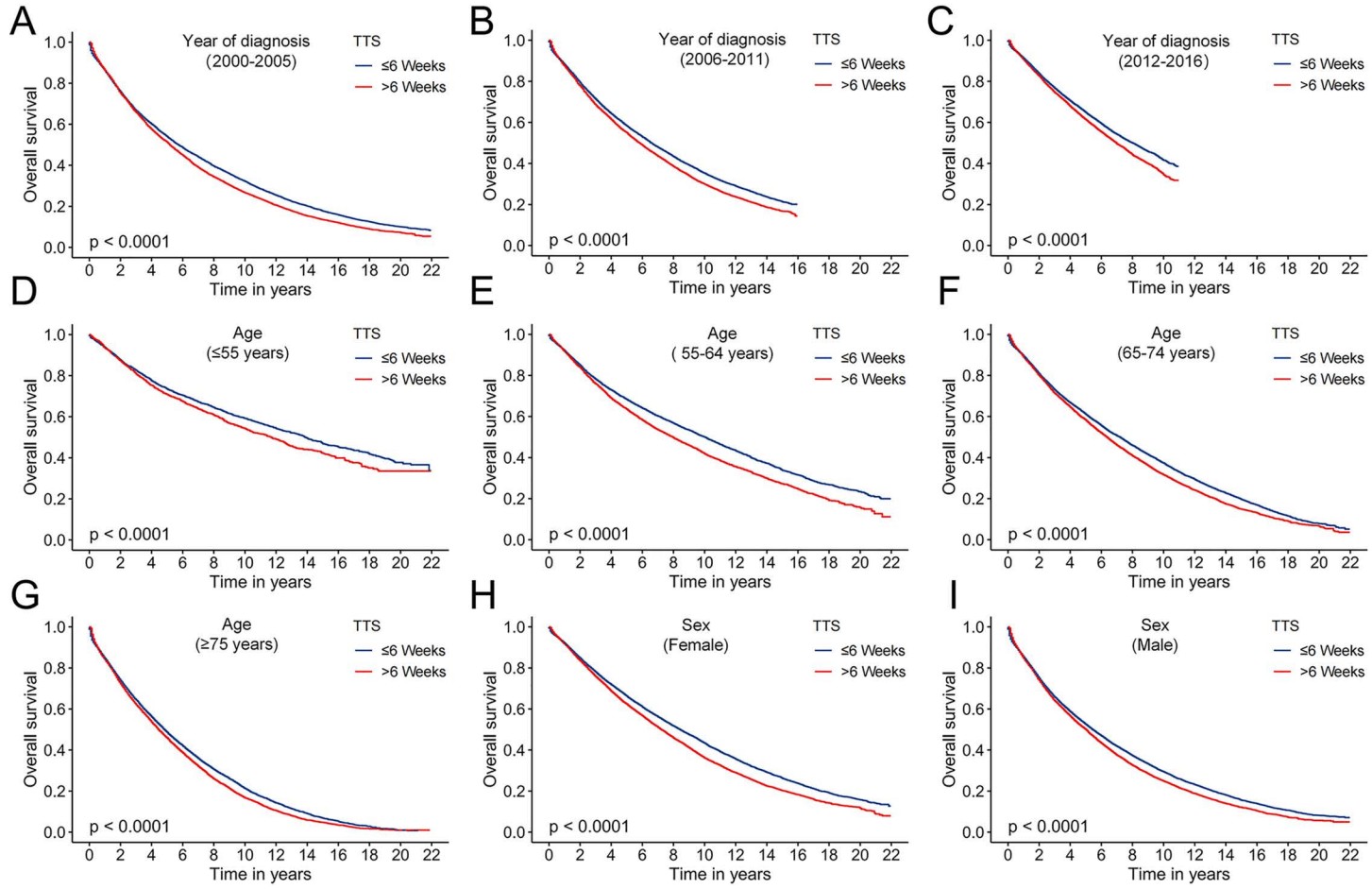

**Fig 4. Kaplan-Meier curves of the TTS ≤6 weeks group and the TTS>6 weeks surgery group across different subgroups.** (A) year of diagnosis (2000-2005), (B) year of diagnosis (2006-2011), (C) year of diagnosis (2012-2016), (D) age (< 55 years), (E) (55-64 years), (F) age (65-74 years), (G) age (≥ 75 years), (H) sex (female) and (I) sex (male).

example, patients with severe comorbidities require longer preoperative preparation time but have a poorer prognosis. Third, there is no information about the definitive way of diagnosis (e.g., biopsy or imaging or sputum cytology) and the specific date associated with it, which may affect the urgency with which the patient chooses surgical treatment. Finally, this study lacks an external validation, so the conclusions results cannot be generalized to lung cancer patients from other countries and ethnicities worldwide. In the future, we will conduct a prospective study to validate our results and examine factors that potentially may influence delayed TTS.

## Conclusion

In conclusion, this study demonstrated that stage I-II NSCLC patients who underwent surgery within 6 weeks of diagnosis had significantly higher survival rates than those who underwent surgery after 6 weeks. Fortunately, more than half of the patients underwent surgery within this time frame. Additionally, we identified that age, sex, race, and marital status were independent risk factors for delayed TTS after diagnosis in patients with NSCLC. In future studies, we will

**Table 4. Multinomial logistic regression of 51-75th Percentile (TTS 41-62 days), 75-95th Percentile (TTS 63-111 days) and >95th Percentile (TTS > 111 days) groups compared to a reference group of patients in the 0 to 50th percentile (TTS 1-40 days).**

| Demographic characteristics | OR (95% CI) P value | | | | | |
|---|---|---|---|---|---|---|
| | 51-75th Percentile | | 76-95th Percentile | | > 95th Percentile | |
| | (41–62 days) | | (63–111 days) | | (> 111 days) | |
| **Year of diagnosis** | | | | | | |
| 2000-2005 | 1.00 (Reference) | | 1.00 (Reference) | | 1.00 (Reference) | |
| 2006-2011 | 1.22 (1.16, 1.29) | < 0.001 | 1.24 (1.17, 1.32) | < 0.001 | 1.16 (1.05, 1.27) | 0.002 |
| 2012-2016 | 1.43 (1.36, 1.50) | < 0.001 | 1.39 (1.31, 1.47) | < 0.001 | 1.45 (1.32, 1.58) | < 0.001 |
| **Age** | | | | | | |
| <55 years | 1.00 (Reference) | | 1.00 (Reference) | | 1.00 (Reference) | |
| 55–64 years | 1.20 (1.10, 1.32) | < 0.001 | 1.26 (1.14, 1.40) | < 0.001 | 1.17 (1.01, 1.36) | 0.039 |
| 65–74 years | 1.31 (1.20, 1.43) | < 0.001 | 1.35 (1.22, 1.49) | < 0.001 | 1.13 (0.98, 1.31) | 0.099 |
| ≥75 years | 1.48 (1.36, 1.63) | < 0.001 | 1.46 (1.32, 1.62) | < 0.001 | 1.27 (1.09, 1.47) | 0.002 |
| **Sex** | | | | | | |
| Female | 1.00 (Reference) | | 1.00 (Reference) | | 1.00 (Reference) | |
| Male | 1.04 (1.00, 1.09) | 0.064 | 1.15 (1.09, 1.20) | < 0.001 | 1.32 (1.22, 1.42) | < 0.001 |
| **Race** | | | | | | |
| White | 1.00 (Reference) | | 1.00 (Reference) | | 1.00 (Reference) | |
| Black | 1.23 (1.14, 1.34) | < 0.001 | 1.50 (1.37, 1.63) | < 0.001 | 1.99 (1.77, 2.23) | < 0.001 |
| Other | 1.15 (1.05, 1.25) | 0.002 | 1.32 (1.20, 1.45) | < 0.001 | 1.51 (1.32, 1.73) | < 0.001 |
| Unknown | 0.69 (0.36, 1.33) | 0.268 | 0.84 (0.42, 1.66) | 0.617 | 1.28 (0.53, 3.07) | 0.584 |
| **Marital status** | | | | | | |
| Married | 1.00 (Reference) | | 1.00 (Reference) | | 1.00 (Reference) | |
| Divorced | 1.20 (1.13, 1.28) | < 0.001 | 1.61 (1.50, 1.73) | < 0.001 | 1.82 (1.63, 2.02) | < 0.001 |
| Single (never married) | 1.17 (1.09, 1.26) | < 0.001 | 1.54 (1.43, 1.66) | < 0.001 | 2.05 (1.84, 2.29) | < 0.001 |
| Widow | 1.20 (1.13, 1.28) | < 0.001 | 1.52 (1.42, 1.62) | < 0.001 | 1.56 (1.41, 1.74) | < 0.001 |
| Unknown | 0.92 (0.81, 1.03) | 0.161 | 1.03 (0.90, 1.17) | 0.710 | 1.18 (0.97, 1.45) | 0.101 |

**Abbreviations:** CI, confidence interval; OR, odds ratio.

prospectively follow and investigate patients who experience delayed surgery to identify additional factors influencing delayed TTS.

## Supporting information

**Fig S1. Forest plot of the multivariate Cox regression analysis for overall survival in stage I-II NSCLC.** (TIF)

**Fig S2. Forest plot of the multivariate Cox regression analysis for overall survival in stage I-II NSCLC after IPTW (Stratified by TTS ≤ 6 weeks and > 6 weeks).** (TIF)

**Fig S3. Kaplan–Meier curves of the TTS ≤ 6 weeks group and the TTS > 6 weeks surgery group in different subgroups.** (A) race (White), (B) race (Black), (C) race (other), (D) histologic type (LUAD), (E) histologic type (LSCC), (F) histologic type (other), (G) lymph node positivity (0), (H) lymph node positivity (1–3), (I) lymph node positivity (> 3). (TIF)

**Fig S4. Kaplan–Meier curves of the TTS ≤ 6 weeks group and the TTS > 6 weeks surgery group in different sub-groups.** (A) grade (I), (B) grade (II), (C) grade (III), (D) grade (IV), (E) tumor size (0–3 cm), (F) tumor size (> 3 cm and ≤ 5 cm) and (G) tumor size (> 5 cm), (H) stage (I), (I) stage (II), (J) surgery type (wedge resection), (K) surgery type (segmentectomy) and (L) surgery type (lobectomy).
(TIF)

**Table S1. Characteristics of stage I-II NSCLC patients before and after IPTW in the TTS ≤ 6 weeks and TTS > 6 weeks groups.**
(DOCX)

**Table S2. Multinomial logistic regression of 51–75th Percentile (TTS 41–62 days), 75–95th Percentile (TTS 63–111 days) and > 95th Percentile (TTS > 111 days) groups compared to a reference group of patients in the 0–50th percentile (TTS 1–40 days).**
(DOCX)

## Acknowledgments

The authors would like to extend sincere gratitude to all public health professionals involved in the SEER database.

## Author contributions

**Conceptualization:** Ye Zhang, Yeji Hu, Jinfeng Xi, Bo Wu, Wenxiong Zhang, Chunling Li.

**Data curation:** Ye Zhang, Yeji Hu, Jinfeng Xi, Bo Wu, Wenxiong Zhang, Chunling Li.

**Formal analysis:** Ye Zhang, Yeji Hu, Jinfeng Xi, Bo Wu, Wenxiong Zhang, Chunling Li.

**Investigation:** Ye Zhang, Chunling Li.

**Methodology:** Ye Zhang, Chunling Li.

**Project administration:** Ye Zhang, Chunling Li.

**Resources:** Ye Zhang, Chunling Li.

**Software:** Ye Zhang, Chunling Li.

**Supervision:** Ye Zhang, Chunling Li.

**Validation:** Ye Zhang, Chunling Li.

**Visualization:** Ye Zhang, Chunling Li.

**Writing – original draft:** Ye Zhang, Yeji Hu, Jinfeng Xi, Bo Wu, Wenxiong Zhang, Chunling Li.

**Writing – review & editing:** Ye Zhang, Chunling Li.

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
