## [Decision Letter · Decision Letter 0]

26 Feb 2025

PONE-D-24-46752Critical timing: Impact of delays to surgery on prognosis in stage I-II non-small cell lung cancerPLOS ONE

Dear Dr. Li,

Thank you for submitting your manuscript to PLOS ONE. After careful consideration, we feel that it has merit but does not fully meet PLOS ONE’s publication criteria as it currently stands. Therefore, we invite you to submit a revised version of the manuscript that addresses the points raised during the review process.

Indeed, after a thorough evaluation of your manuscript, we have decided to involve a statistical reviewer to provide additional insights regarding the analysis presented in your study.

We kindly invite you to carefully consider the reviewer’s suggestions and implement the necessary adjustments in the revised version of your manuscript. These improvements will help enhance the clarity and robustness of your findings.

We look forward to receiving your revised manuscript.

Kind regards,

Francesco Guerrera, M.D., Ph.D.

Academic Editor

PLOS ONE

Journal Requirements:

Reviewers' comments:

Reviewer's Responses to Questions

**Comments to the Author**

1. Is the manuscript technically sound, and do the data support the conclusions?

Reviewer #1: Yes

Reviewer #2: Yes

Reviewer #3: Yes

2. Has the statistical analysis been performed appropriately and rigorously? 

Reviewer #1: Yes

Reviewer #2: I Don't Know

Reviewer #3: No

3. Have the authors made all data underlying the findings in their manuscript fully available?

Reviewer #1: Yes

Reviewer #2: Yes

Reviewer #3: Yes

4. Is the manuscript presented in an intelligible fashion and written in standard English?

Reviewer #1: Yes

Reviewer #2: Yes

Reviewer #3: Yes

5. Review Comments to the Author

Reviewer #1: I enjoyed reading the manuscript: "Critical timing: Impact of delays to surgery on prognosis in stage I-II non-small cell lung cancer". I found that the manuscript is very well written, the methodology is sound and the findings are relevant.

I have only minor comments:

1) On page 9, lines 164-166: "OS was significantly better in the group with a TTS of 7 weeks compared to the earliest surgery group (HR: 1.06, 95% CI: 1.02-1.10, P = 0.002)." Isn't it the opposite? I understood that survival was worse in patients with longer TTS

2) On page 11 lines 202-204, median TTS are reported as pure numbers, those number represent days, so I suggest to add the unit of measurement there.

3) In the conclusion, page 14 lines 285-286: "Unfortunately, most patients underwent surgery within this time frame". I don't understand this sentence. Why would it be unfortunate that most patients be operated on within 6 weeks from diagnosis if this is associated with better survival? Besides, by doing some simple math from table 2, I calculated that 54% patients underwent surgery within 6 weeks. So this is roughly half patients' rather than "most" patients.

4) I would like the authors to provide a rationale that justifies the hypothesis that longer time to surgery results in worse survival. Do 7, 14 or 21 more days really matter? What would be the explanation for this? Cancer progression? Isn't it rather the case that TTS correlates with some unmeasured confounder?

Reviewer #2: The authors reported an interesting study on the risk factors for delayed time to surgery based on the SEER database. The article is very interesting and it is well written.

I have only few comments:

- did the authors analyzed the rate of pathological upstaging in patients with a delayed (more than 6 weeks) TTS?

- English language should be reviewed and some minor typos or phrase constructions should be checked.

Reviewer #3: The study investigates the relationship between time to surgery (TTS) and overall survival (OS) in stage I-II NSCLC using data from the SEER database (2000–2016). While the research question is clinically important, several methodological and statistical aspects require further clarification to ensure the robustness of the findings.

Major Methodological and Statistical Concerns

1. Definition and Uniformity of Study Start Date

• How was the date of diagnosis consistently defined across all patients, given potential variability in hospital records?

• Were there protocols to ensure uniformity in determining the start of TTS (e.g., using biopsy dates or first imaging confirmation)?

2. Potential Selection Bias and Confounding

• How was selection bias addressed, particularly since earlier-surgery patients may be inherently healthier than those with delays?

• Was propensity score matching or inverse probability weighting (IPTW) considered to account for baseline differences between groups?

• Were any hospital or healthcare access factors (e.g., treatment facility volume) considered in the analysis?

3. Temporal Changes in Treatment (2000–2016) and Impact on Survival

• How does the study account for advancements in treatment over the 16-year study period?

• Were patients diagnosed in earlier years at a disadvantage compared to those diagnosed later, due to improved surgical techniques, perioperative care, and adjuvant therapies?

• Was year of diagnosis included as a variable in the Cox regression models to adjust for potential survival improvements over time?

• Would a stratified analysis by treatment era (e.g., 2000–2005, 2006–2011, 2012–2016) help in distinguishing whether observed survival differences are due to TTS or treatment advancements?

4. Justification for the 6-Week Threshold for Delayed Surgery

• What was the rationale for defining surgical delay as >6 weeks?

• Was this threshold based on prior studies, clinical guidelines, or data-driven statistical cutoffs?

• Would different cut-off points (e.g., 4 weeks, 8 weeks) yield different conclusions?

5. Handling of Time-Dependent Selection Bias (Landmark or Time-Dependent Cox Model?)

• How was time-dependent selection bias addressed, given that patients who undergo early surgery must have survived long enough to receive surgery (immortal time bias)?

• Did the authors consider a landmark analysis, where survival is analyzed only for patients still at risk after a set time point (e.g., 6 weeks)?

• If not, was a time-dependent Cox model considered to model surgery as a time-varying covariate?

6. Handling of Missing Data

• How were missing values handled, particularly for key variables such as:

o Lymph node status (9.67% missing)

o Tumor size (0.80% missing)

o Grade (6.55% missing)

• Was multiple imputation performed, or were missing cases excluded?

• Were sensitivity analyses conducted to assess the impact of missing data on the results?

7. Statistical Assumptions in Survival Models

• Was the proportional hazards assumption tested in the Cox regression models (e.g., using Schoenfeld residuals)?

• If the assumption was violated, were alternative approaches (e.g., stratified Cox models, time-varying covariates) considered?

• Were log-rank tests formally reported alongside Kaplan-Meier survival curves?

8. Multivariable Regression and Collinearity

• How was collinearity between variables (e.g., age, comorbidities, socioeconomic factors) assessed in the multivariate models?

• Were Variance Inflation Factors (VIFs) checked to ensure stability of the regression models?

9. Potential Reverse Causation

• Could reverse causation explain the association between delayed surgery and worse survival (i.e., patients with worse prognosis may have had surgery delayed due to complications or workup delays)?

• Was a sensitivity analysis performed to examine whether delays were more common in patients with high-risk disease features?

Minor Comments

1. Clarification of Data Availability

o How can readers replicate the study? Was the specific SEER dataset version and variable set documented?

2. Presentation of Results

o Are all tables and figures self-explanatory? Were abbreviations and statistical measures clearly defined?

o Would a forest plot of hazard ratios improve clarity in multivariable analyses?

3. Generalizability of Findings

o Given that the study is based on U.S.-based SEER data, can the results be generalized to other healthcare settings with different surgical wait times?

6. PLOS authors have the option to publish the peer review history of their article (what does this mean? ). If published, this will include your full peer review and any attached files.

**Do you want your identity to be public for this peer review?** For information about this choice, including consent withdrawal, please see our Privacy Policy .

Reviewer #1: **Yes: ** Marco Mammana

Reviewer #2: **Yes: ** Pietro Bertoglio

Reviewer #3: No

---

## [Author Response · Author response to Decision Letter 0]

1 Apr 2025

Dear Editors and reviewers:

Thank you for your letter and the reviewers' comments concerning our manuscript entitled “Critical timing: Impact of delays to surgery on prognosis in stage I-II non-small cell lung cancer” to PLOS ONE (ID: PONE-D-24-46752). We appreciate the opportunity to revise and improve our work according to the journal's standards. The reviewers' suggestions have proven invaluable for enhancing the quality of our manuscript. We have meticulously addressed each comment through a point-by-point response. All modifications in the manuscript have been explicitly highlighted in red text for editorial review. The main corrections in the paper and the responds to the reviewers' comments are as following:

Response to the Editor’s comments:

1. Comment: Thank you for submitting your manuscript to PLOS ONE. After careful consideration, we feel that it has merit but does not fully meet PLOS ONE’s publication criteria as it currently stands. Therefore, we invite you to submit a revised version of the manuscript that addresses the points raised during the review process.

Indeed, after a thorough evaluation of your manuscript, we have decided to involve a statistical reviewer to provide additional insights regarding the analysis presented in your study.

We kindly invite you to carefully consider the reviewer’s suggestions and implement the necessary adjustments in the revised version of your manuscript. These improvements will help enhance the clarity and robustness of your findings.

Response and changes: Thank you very much for your careful reading of my manuscript and your valuable comments. As you mentioned, there are some shortcomings in the statistical section of the manuscript. The expert's statistical review provided valuable insights, which will be of great help in improving the quality of the manuscript. We will make revisions one by one according to the reviewers' suggestions.

Response to Reviewer #1's comments

1. Comment: I enjoyed reading the manuscript: "Critical timing: Impact of delays to surgery on prognosis in stage I-II non-small cell lung cancer". I found that the manuscript is very well written, the methodology is sound and the findings are relevant.

Response and changes: Thank you very much for your valuable comments and recognition of our manuscript. We are pleased to hear that you find the manuscript well written, with sound methodology and relevant findings. Your encouragement is greatly appreciated, and we will revise the manuscript based on your valuable suggestions and work hard to further improve its quality.

2. Comment: 1) On page 9, lines 164-166: "OS was significantly better in the group with a TTS of 7 weeks compared to the earliest surgery group (HR: 1.06, 95% CI: 1.02-1.10, P = 0.002)." Isn't it the opposite? I understood that survival was worse in patients with longer TTS

Response and changes: Thank you very much for your careful reading and valuable comments. I am embarrassed by my oversight, and as you mentioned, the survival was worse in patients with a longer TTS. I mistakenly wrote that survival was better in the group with a TTS of 7 weeks. Thank you for your correction. The authors have revised this part of the expression accordingly. (see details on page 9, line 167 to line 168 of the revised manuscript):

Original:

"OS was significantly better in the group with a TTS of 7 weeks compared to the earliest surgery group (HR: 1.06, 95% CI: 1.02-1.10, P = 0.002)."

Revised:

"OS was significantly worse in the group with a TTS of 7 weeks compared to the earliest surgery group (HR: 1.08, 95% CI: 1.04-1.12, P < 0.001)."

3. Comment: 2) On page 11 lines 202-204, median TTS are reported as pure numbers, those number represent days, so I suggest to add the unit of measurement there.

Response and changes: Thank you for your valuable comments. The median TTS in our manuscript was expressed using pure numbers without specifying the units, which was not sufficiently rigorous and did not adhere to academic writing standards. Following your suggestion, the authors have now added the appropriate units after the corresponding numbers. (see details on page 11, line 208 to line 2011 of the revised manuscript):

Original:

"Compared to the median (IQR) of TTS for lung wedge resections (43 (26, 69)), the median (IQR) of TTS for resections of segmentectomy (41 (24, 66), P = 0.002) and lobectomy (40 (25, 62), P<0.001) was significantly shorter."

Revised:

"Compared to the median (IQR) of TTS for lung wedge resections (43 (26, 69) days), the median (IQR) of TTS for resections of segmentectomy (41 (24, 66) days, P = 0.002) and lobectomy (40 (25, 62) days, P < 0.001) was significantly shorter."

4. Comment: 3) In the conclusion, page 14 lines 285-286: "Unfortunately, most patients underwent surgery within this time frame". I don't understand this sentence. Why would it be unfortunate that most patients be operated on within 6 weeks from diagnosis if this is associated with better survival? Besides, by doing some simple math from table 2, I calculated that 54% patients underwent surgery within 6 weeks. So this is roughly half patients' rather than "most" patients.

Response and changes: Thank you for your valuable comments. After carefully reviewing the manuscript, we realized that we made an error in our expression. We should have written "Fortunately" but mistakenly wrote "Unfortunately." Furthermore, regarding your point about 54% not being considered "most," we now understand that "most" typically refers to a percentage greater than 50%, and while 54% is slightly above 50%, it doesn't clearly convey the idea of "most." Using "most" does not align with the rigor and accuracy required in academic writing, and terms like "more than half" or "about half" are more appropriate. Based on your suggestion, the authors have revised this sentence accordingly. (see details on page 15, line 297 to line 298 of the revised manuscript):

Original:

"Unfortunately, most patients underwent surgery within this time frame."

Revised:

"Fortunately, more than half of the patients underwent surgery within this time frame."

5. Comment: 4) I would like the authors to provide a rationale that justifies the hypothesis that longer time to surgery results in worse survival. Do 7, 14 or 21 more days really matter? What would be the explanation for this? Cancer progression? Isn't it rather the case that TTS correlates with some unmeasured confounder?

Response and changes: Thank you for your valuable comments. Regarding the relationship between surgical delay and survival, our hypothesis suggests that surgical delay may lead to further tumor progression, increasing tumor burden, and thus raising the patient's risk of mortality, which results in a lower survival rate. As for delays of 7, 14, or 21 days, although the delay duration is relatively short, it could still impact tumor development, especially for malignant tumors. Additionally, TTS is indeed associated with certain unmeasured confounding factors. For example, the patient's overall health status, comorbidities, and available medical resources may influence the timing of surgery. Therefore, surgical delay may be related not only to tumor progression but also to interactions with these unmeasured factors. However, our study has made efforts to control for available confounders to ensure the reliability of the conclusions. Due to the limitations of the SEER database, relevant information such as patients' underlying conditions was not recorded. In the future, we will conduct prospective studies to validate the results and explore potential factors influencing delayed TTS. The authors have addressed these points in the discussion and limitations sections. (see details on page 14, line 279 to line 283 of the revised manuscript):

The following was added:

“Second, the SEER database lacks comprehensive information regarding significant comorbidities, which may affect the accuracy of our findings. For example, patients with severe comorbidities require longer preoperative preparation time but have a poorer prognosis.

In the future, we will conduct a prospective study to validate our results and examine factors that potentially may influence delayed TTS.

Response to Reviewer #2's comments

1. Comment: The authors reported an interesting study on the risk factors for delayed time to surgery based on the SEER database. The article is very interesting and it is well written.

Response and changes: Thank you for your recognition and valuable comments on our manuscript. We are honored that you find our study interesting and well-written. We will further improve the manuscript based on your suggestions to ensure its quality continues to rise.

2. Comment: did the authors analyzed the rate of pathological upstaging in patients with a delayed (more than 6 weeks) TTS?

Response and changes: Thank you very much for your comments. Since all the patients included in our study underwent surgery, all of our tumor staging is based on pathological staging. Due to the limitations of the SEER database, we did not find information that provides pathological staging after clinical staging. The patients with clinical staging are those who neither underwent surgery nor had lung tissue or lymph node biopsies, and were diagnosed solely based on imaging. Therefore, I did not analyze the rate of pathological upstaging in patients who underwent surgery more than 6 weeks after diagnosis. Based purely on pathological staging, there was no significant difference in tumor pathological staging between patients who underwent surgery within 6 weeks and those who underwent surgery after 6 weeks.

3. Comment: English language should be reviewed and some minor typos or phrase constructions should be checked.

Response and changes: Thank you for your valuable comments. The authors have submitted the manuscript to a proofreading company for further editing, making necessary revisions to correct minor typos or phrase constructions in order to improve the accuracy and clarity of the language.

Response to Reviewer #3's comments

1. Comment: The study investigates the relationship between time to surgery (TTS) and overall survival (OS) in stage I-II NSCLC using data from the SEER database (2000-2016). While the research question is clinically important, several methodological and statistical aspects require further clarification to ensure the robustness of the findings.

Response and changes: Thank you for your valuable comments and for carefully reviewing the methodology of the manuscript. We appreciate your recognition of the clinical importance of our research question. We acknowledge that further clarification is needed regarding the methodological and statistical aspects to ensure the robustness of our findings. We have carefully reviewed the study design, analysis methods, and statistical sections. We will revise the corresponding sections based on your comments. Each of your suggestions has been very helpful in improving the quality of our manuscript, and we value your insights. I will also take this opportunity to further study and consolidate my knowledge in statistics and methodology to ensure higher quality papers in the future.

2. Comment: 1. Definition and Uniformity of Study Start Date

• How was the date of diagnosis consistently defined across all patients, given potential variability in hospital records?

• Were there protocols to ensure uniformity in determining the start of TTS (e.g., using biopsy dates or first imaging confirmation)?

Response and changes: Thank you for your valuable comments on our manuscript. All of our data is sourced from the SEER database, which was initiated by the National Cancer Institute (NCI) in 1973. The SEER database aims to provide a comprehensive understanding of cancer epidemiology, treatment outcomes, and survival rates by collecting and analyzing cancer case data. It marks the beginning of standardized and systematic collection of cancer statistics. The cancer information submitted by hospitals follows a unified standard. However, specific information related to TTS is not accessible to us. Moreover, TTS is labeled in the database as "Time from diagnosis to treatment in days," with the definition being "Recode Number of days from diagnosis to date of initial treatment, calculated using complete dates." The SEER database does not specify whether the diagnosis date refers to the biopsy date or the date of initial imaging confirmation. This may impact the study results. In future prospective studies, we will ensure a clear definition of whether the diagnosis date refers to the biopsy date or the date of initial imaging confirmation. The authors will discuss in the limitations section. (see details on page 14, line 286 to line 288 of the revised manuscript):

The following was added:

"Third, there is no information about the definitive way of diagnosis (e.g., biopsy or imaging or sputum cytology) and the specific date associated with it, which may affect the urgency with which the patient chooses surgical treatment."

3. Comment: Potential Selection Bias and Confounding

• How was selection bias addressed, particularly since earlier-surgery patients may be inherently healthier than those with delays?

• Was propensity score matching or inverse probability weighting (IPTW) considered to account for baseline differences between groups?

• Were any hospital or healthcare access factors (e.g., treatment facility volume) considered in the analysis?

Response and changes: Thank you very much for your valuable comments. We sincerely apologize for not considering the bias of healthier patients undergoing early surgery. As shown in Table S1 of the manuscript, there are significant differences between the patients who underwent surgery within six weeks and those who had surgery after six weeks in several variables. To better adjust for selection bias and confounding factors. We first did propensity matching analysis (PSM), but the method could not could not balance the baseline characteristics between the two groups well. So we chose inverse probability of treatment weighting (IPTW) to control for confounding variables between the groups, reduce selection bias, and enhance the credibility of our observational study. Furthermore, medical institution resources (such as medical equipment, supplies, physician expertise, and surgical volume) may influence the prognosis of our lung cancer postoperative patients. However, since the data for this study comes from the SEER database, we are unable to access relevant information regarding these factors. We hope to incorporate more information related to delayed surgery and prognosis in future studies to further validate our findings. The results of the propensity score matching analysis can be found in Table S1, and the corresponding figures and tables in the manuscript have been updated accordingly.

Update table S1and table 3

4. Comment: Temporal Changes in Treatment (2000–2016) and Impact on Survival

• How does the study account for advancements in treatment over the 16-year study period?

• Were patients diagnosed in earlier years at a disadvantage compared to those diagnosed later, due to improved surgical techniques, perioperative care, and adjuvant therapies?

• Was year of diagnosis included as a variable in the Cox regression models to adjust for potential survival improvements over time?

• Would a stratified analysis by treatment era (e.g., 2000–2005, 2006–2011, 2012–2016) help in distinguishing whether observed survival differences are due to TTS or treatment advancements?

Response and changes: Thank you very much for your valuable comments. We sincerely apologize for overlooking the impact of changes in treatment methods over time on this study. The Cox model did not include the year of diagnosis. Following your suggestion, we have re-downloaded the data, incorporated the year as an important variable, and performed stratification by the periods 2000–2005, 2006–2011, and 2012–2016. The corresponding figures, tables, and data in the manuscript have been updated accordingly.

Updated all tables and figure 3.

Figure 4 The Kaplan‒Meier curves of the TTS ≤ 6 weeks group and the TTS > 6 weeks su

---

## [Decision Letter · Decision Letter 1]

22 Apr 2025

Critical timing: Impact of delays to surgery on prognosis in stage I-II non-small cell lung cancer

PONE-D-24-46752R1

Dear Dr. Li,

We’re pleased to inform you that your manuscript has been judged scientifically suitable for publication and will be formally accepted for publication once it meets all outstanding technical requirements.

Kind regards,

Francesco Guerrera, M.D., Ph.D.

Academic Editor

PLOS ONE

Additional Editor Comments (optional):

Reviewers' comments:

Reviewer's Responses to Questions

**Comments to the Author**

1. If the authors have adequately addressed your comments raised in a previous round of review and you feel that this manuscript is now acceptable for publication, you may indicate that here to bypass the “Comments to the Author” section, enter your conflict of interest statement in the “Confidential to Editor” section, and submit your "Accept" recommendation.

Reviewer #3: All comments have been addressed

2. Is the manuscript technically sound, and do the data support the conclusions?

Reviewer #3: Yes

3. Has the statistical analysis been performed appropriately and rigorously? 

Reviewer #3: Yes

4. Have the authors made all data underlying the findings in their manuscript fully available?

Reviewer #3: Yes

5. Is the manuscript presented in an intelligible fashion and written in standard English?

Reviewer #3: Yes

6. Review Comments to the Author

Reviewer #3: (No Response)

7. PLOS authors have the option to publish the peer review history of their article (what does this mean? ). If published, this will include your full peer review and any attached files.

**Do you want your identity to be public for this peer review?** For information about this choice, including consent withdrawal, please see our Privacy Policy .

Reviewer #3: No

---

## [Editor Report · Acceptance letter]

PONE-D-24-46752R1

PLOS ONE

Dear Dr. Li,

I'm pleased to inform you that your manuscript has been deemed suitable for publication in PLOS ONE. Congratulations! Your manuscript is now being handed over to our production team.

Kind regards,

on behalf of

Dr. Francesco Guerrera

Academic Editor

PLOS ONE